# Applying Physics-Informed Neural Networks to Solve Navier–Stokes Equations for Laminar Flow around a Particle

**Beichao Hu**  **and Dwayne McDaniel ***

Department of Mechanical and Materials Engineering, Florida International University, Miami, FL 33199, USA; bhu007@fiu.edu
* Correspondence: mcdaniel@fiu.edu

**Abstract:** In recent years, Physics-Informed Neural Networks (PINNs) have drawn great interest among researchers as a tool to solve computational physics problems. Unlike conventional neural networks, which are black-box models that "blindly" establish a correlation between input and output variables using a large quantity of labeled data, PINNs directly embed physical laws (primarily partial differential equations) within the loss function of neural networks. By minimizing the loss function, this approach allows the output variables to automatically satisfy physical equations without the need for labeled data. The Navier–Stokes equation is one of the most classic governing equations in thermal fluid engineering. This study constructs a PINN to solve the Navier–Stokes equations for a 2D incompressible laminar flow problem. Flows passing around a 2D circular particle are chosen as the benchmark case, and an elliptical particle is also examined to enrich the research. The velocity and pressure fields are predicted by the PINNs, and the results are compared with those derived from Computational Fluid Dynamics (CFD). Additionally, the particle drag force coefficient is calculated to quantify the discrepancy in the results of the PINNs as compared to CFD outcomes. The drag coefficient maintained an error within 10% across all test scenarios.

**Keywords:** Navier–Stokes equations; physics-informed neural networks; CFD; particle

## 1. Introduction

In the past few decades, numerical simulations have been commonly considered one of the most effective methods for solving non-linear Partial Differential Equations (PDEs) in most engineering problems. One such classical problem is the solution of Navier–Stokes equations through Computational Fluid Dynamics (CFD). However, numerical simulations can be computationally prohibitive due to the necessity of generating a large-scale and intricate computational grid, a complex process that requires a high level of expertise and specialized domain knowledge in many real-world applications. In recent years, there has been a surge in efforts to apply new techniques to solve these conventional engineering problems, with deep learning emerging as a prominent approach, thanks to a number of factors such as advancement of training algorithms, new architectures and techniques, and the leap in computing power. Deep learning excels at approximating complex relations between input and output variables through elementary operations implemented by an artificial neural network (ANN). A loss function, typically constructed as the mean squared error between the predictions made by the ANN and the ground truth (labeled data), is minimized (ideally to zero) through optimization algorithms. This allows the ANN to predict results close to the ground truth. Deep learning has proven to be powerful in tackling multi-scale and non-linear problems, especially when abundant observational data is available. It is unsurprising that deep learning has achieved considerable success in conventional engineering problems, where the mathematical relationships between inputs and outputs are often unclear and analytical solutions are non-existent.

Despite its potency and versatility in addressing numerous engineering problems, deep learning comes with several downsides. First, it requires a vast amount of labeled data,

which can be prohibitively expensive for most engineering problems. Secondly, the trained neural network is a "black box" that is difficult to interpret and not directly transferrable to human knowledge. Thirdly, most deep learning processes in engineering problems are characterized as "data rich, knowledge poor" [1], as the models trained are predominantly based on labeled data with no consideration of physical law constraints. Consequently, there is a pressing need to incorporate fundamental physical laws and domain knowledge into deep learning approaches [2].

In recent years, a novel category of deep learning known as Physics Informed Neural Networks (PINNs) has emerged. PINNs essentially alter the loss function from being a measure of the discrepancy between predictions and labeled data to being the residuals of governing equations, primarily PDEs. Through the minimization of these PDE residuals, the trained ANN inherently conforms to the governing equations, accurately reflecting physical laws. Furthermore, by similarly constraining the boundary and initial conditions, there is no requirement for labeled data during the training process.

Preliminary studies on PINNs began around 2018 as proof-of-concept studies. Raissi et al. [3] introduced the foundational framework of PINNs and applied it to solve several PDEs, including Burger's equation, the Korteweg de Vries (KdV) equation, and the Kuramoto–Sivashinsky equation. Automatic Differentiation (AD) was used to calculate the partial derivates in PDEs, with the PDE residual calculated as the loss function to be minimized. Boundary conditions and initial conditions were softly constrained within the loss function to ensure a unique solution upon convergence. Since then, several PINN frameworks have been proposed to solve various PDEs, to name a few [4–10].

In the specific context of Navier–Stokes equations, several studies have made significant contributions. Sun et al. [11] proposed a surrogate model to solve incompressible Navier–Stokes equations. Boundary and initial conditions were enforced in a "hard" way using an encoder-decoder structure. Jin et al. [12] proposed the NSF net to solve three-dimensional Navier–Stokes equations in both laminar and turbulent flow regimes. Rao et al. [13] proposed a framework to solve mixed-form incompressible Navier–Stokes equations that utilized Cauchy stress and the stream function. Gao et al. [14] proposed PhyGeoNet using a Convolutional Neural Network (CNN) to solve finitely differencing discretized Navier–Stokes equations. Ranade et al. [15] proposed the discretization net, which constructed the loss function through a finite volume method similar to the one commonly used in CFD. The partial derivatives were approximated through the Green–Gauss cell-based theorem. Incompressible three-dimensional Navier–Stokes equations, including the energy equation, were solved through the proposed framework.

This study focuses on developing a PINN framework to solve two-dimensional, steady-state, incompressible Navier–Stokes equations. The flow passing around 2D circular and elliptical particles has been chosen as the benchmark case for the validation of the framework. Moreover, the work delves deeper into the application aspect by estimating the particle drag coefficient at several Reynolds (*Re*) numbers. The predicted drag coefficient is then validated against the CFD results. Given the point-based nature of the PINN framework, this study utilized a finite difference method that is different from CFD to discretize the predicted flow field in order to calculate the drag coefficient.

## 2. Methods

### 2.1. Navier–Stokes Equations

In this section, we discuss the methodology for solving the two-dimensional, steady-state, incompressible, laminar Navier–Stokes equations. The equations consist of the mass conservation equation, x-direction momentum equation, and y-direction momentum equation, each presented in their PDE form, as shown in Equations (1)–(3).

$$\frac{\partial u}{\partial x} + \frac{\partial v}{\partial y} = 0 \tag{1}$$

$$-\frac{\partial p}{\partial x} + \mu\left(\frac{\partial^2 u}{\partial x^2} + \frac{\partial^2 u}{\partial y^2}\right) = \rho\left(u\frac{\partial u}{\partial x} + v\frac{\partial u}{\partial y}\right) \tag{2}$$

$$-\frac{\partial p}{\partial y} + \mu\left(\frac{\partial^2 v}{\partial x^2} + \frac{\partial^2 v}{\partial y^2}\right) = \rho\left(u\frac{\partial v}{\partial x} + v\frac{\partial v}{\partial y}\right) \tag{3}$$

where $u$ and $v$ are the velocity components in $x$ and $y$ directions, respectively, $\rho$ is the density of the fluid, $p$ is the static pressure, and $\mu$ is the dynamic viscosity of the fluid. The exact analytical solution to the above equations is unknown; however, the solution can be formulated such that the transport scalar variables $u$, $v$, and $p$ are a function of the spatial variables $x$ and $y$ (and time variable $t$, should it be a non-steady-state problem).

One of the notable features of a neural network is its ability to approximate nearly any unknown function by stacking elementary operations using the input variables $X$, weights $W$, and bias $b$. As a result, the predicted solution to Navier–Stokes equations (transport scalars $u$, $v$, and $p$) can be formulated as a function of input variables, coupled with weights and bias terms, as shown in Equation (4).

$$[\hat{u}, \hat{v}, \hat{p}] = h_{W,b}(X) = \Phi(WX + b) \tag{4}$$

where $\Phi$ is the activation function of the neural network and $X$ is the input variables $X = [x, y]$.

### 2.2. Automatic Differentiation

The derivatives of the output variables are calculated using Automatic Differentiation (AD) [16]. AD computes derivatives by evaluating the trace of calculations. Since all numerical computations are ultimately compositions of a finite set of elementary operations (e.g., summation and multiplication), the derivatives of which are known [17], the derivative of a complex mathematical equation can be calculated by propagating the derivative of each elementary operation using the chain rule. In the context of PINNs, each elementary operation is recorded in a computational graph when the output variable is constructed by the neural network. A reverse mode of the AD is applied to obtain the derivative of any constructed variables. AD is also utilized during the optimization of the loss function.

It should be noted that in CFD, derivatives are numerically approximated, and the accuracy is influenced by the order of the scheme and the resolution of the grid. Conversely, the derivative computed in PINNs is the exact solution to the mathematical formulation constructed by neural networks, making their accuracy independent of the grid resolution. In essence, PINNs bear resemblance to meshless CFD methods, offering a significant advantage over traditional CFD techniques as they bypass the often complex and time-consuming process of grid generation encountered in many real-world applications.

### 2.3. PINNs Framework

The structure of the 2D steady-state, laminar PINNs framework utilized in this study is depicted in Figure 1. A fully connected neural network, which comprises an input layer housing two neurons, eight hidden layers each containing 60 neurons, and an output layer equipped with three neurons, is employed. The total number of trainable degrees of freedom is 26,105. The input feature consists of the x and y coordinates, while the output feature includes the x-velocity $u$, y-velocity $v$, and static pressure $p$.

The principal difference between PINNs and a traditional ANN is the alteration of the loss function, which usually calculates the mean square error (MSE) between predictions and labeled data and is replaced here by governing equations. In the context of fluid engineering, the Navier–Stokes equations are the governing equations. The neural network predicts variables $u$, $v$, $p$, and their derivatives, which are then incorporated into the mass conservation equation and the x and y momentum equation

(Equations (1)–(3)). The residual of these equations comprises the loss function, denoted as $J_{NS}$ in Equation (5), which represents the aggregate of residuals across all training points in the computational domain.

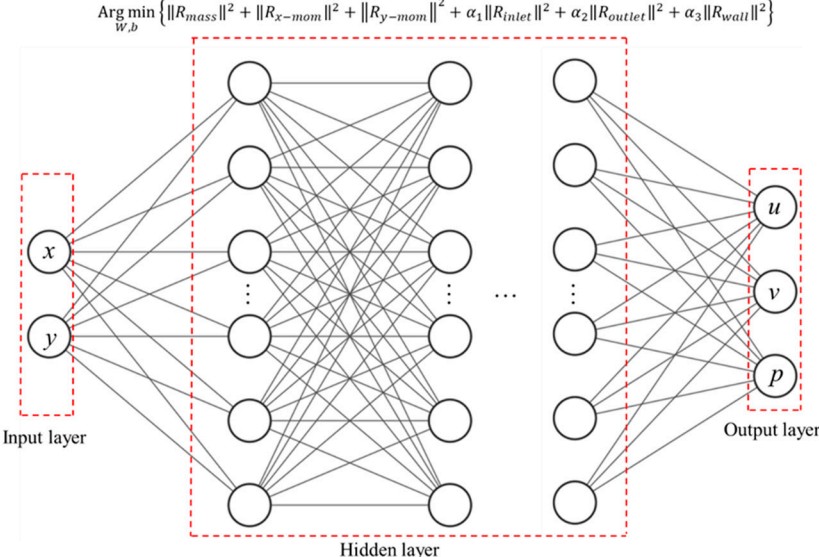

**Figure 1.** PINNs framework proposed in this work.

$$J_{NS} = \alpha_{mass} \frac{1}{N_p} \sum_{i=1}^{N_p} \left\| R_{mass}\left(X^i, W^i, b^i\right) \right\|^2 + \alpha_{x-mom} \frac{1}{N_p} \sum_{i=1}^{N_p} \left\| R_{x-mom}\left(X^i, W^i, b^i\right) \right\|^2$$
$$+ \alpha_{y-mom} \frac{1}{N_p} \sum_{i=1}^{N_p} \left\| R_{y-mom}\left(X^i, W^i, b^i\right) \right\|^2 \tag{5}$$

where $J_{NS}$ is the loss function of Navier–Stokes equations. $R_{mass}$, $R_{x\text{-}mom}$, and $R_{y\text{-}mom}$ are the residuals of Equations (1)–(3), respectively (mass conservation, x-momentum, and y-momentum equations). $N_p$ is the total number of training points in the fluid domain. The residual is calculated as the $l_2$ norm of each equation, as expressed in Equation (6).

$$\|x\|_2 = \sqrt{\sum_{k=1}^{n} x_k^2} \tag{6}$$

here $\overline{x} = [x_1, x_2 \cdots, x_k]^T$.

When the Navier–Stokes loss function $J_{NS}$ is minimized (ideally to zero), each predicted solution at every individual point within the fluid domain complies with the governing equations. Consequently, the predicted values of $u$, $v$, and $p$ are one possible set within an infinite array of solutions to the PDE.

To obtain a unique solution, boundary conditions (and initial conditions in the case of an unsteady-state problem) must be enforced. This is executed similarly to the application of the Navier–Stokes equations, where the boundary conditions essentially form another set of equations within the loss function that needs to be minimized, as illustrated in Equation (7):

$$J_{BC} = \alpha_{Inlet} \frac{1}{N_{pI}} \sum_{i=1}^{N_{pI}} \left\| R_{inlet}\left(X^i, W^i, b^i\right) \right\|^2 + \alpha_{Outlet} \frac{1}{N_{pO}} \sum_{i=1}^{N_{pO}} \left\| R_{outlet}\left(X^i, W^i, b^i\right) \right\|^2$$
$$+ \alpha_{Wall} \frac{1}{N_{pW}} \sum_{i=1}^{N_{pW}} \left\| R_{wall}\left(X^i, W^i, b^i\right) \right\|^2 \tag{7}$$

where $N_{\mathrm{pI}}$, $N_{\mathrm{pO}}$, and $N_{\mathrm{pW}}$ are the number of boundary points on the inlet, outlet, and wall. $R_{\mathrm{inlet}}$, $R_{\mathrm{outlet}}$, and $R_{\mathrm{wall}}$ are the residuals of inlet, outlet, and wall boundary conditions. The total loss function $J_{\mathrm{total}}$ is the summation of both Navier–Stokes loss and the boundary conditions loss, as shown in Equation (8):

$$J_{total} = J_{NS} + J_{BC} \tag{8}$$

$\alpha = \left[\alpha_{\mathrm{mass}}, \alpha_{x-\mathrm{mom}}, \alpha_{y-\mathrm{mom}}, \alpha_{\mathrm{Inlet}}, \alpha_{\mathrm{Outlet}}, \alpha_{\mathrm{wall}}\right]$ is a set of penalizing factors to boost the stability and convergence of the training process. The boundary enforcement on the wall is crucial for mass conservation of the entire flow field, but the sample points on the wall only account for less than 5% of the total points. Therefore, $\alpha_{wall}$ is assigned a value of 2 to compensate for the imbalanced distribution of sample points. For all other terms, $\alpha$ is set to 1.

The neural network is trained by the Adam algorithm [18] for 10,000 iterations and then by the L-BFGS [19] algorithm to improve convergence. The training will stop early when the gradient of each term is less than $1 \times 10^{-7}$. The framework is coded using the Pytorch library.

### 2.4. Investigated Domain

A classical scenario where flow around a 2D circular particle is used as the benchmark to test this framework, as shown in Figure 2.

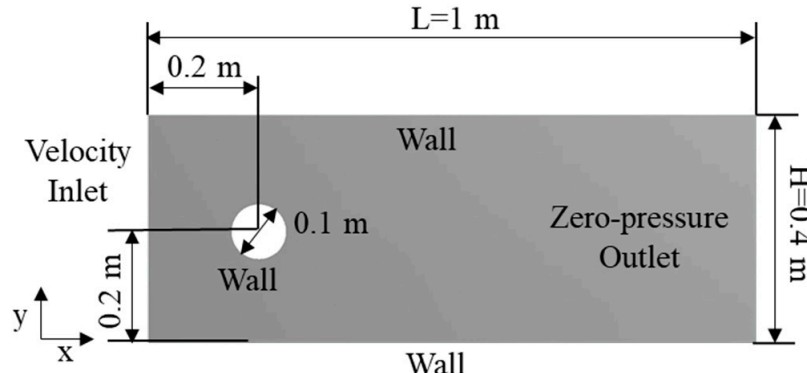

**Figure 2.** Computational domain of the benchmark case.

A parabolic velocity profile is defined as $V = (u, v)$ at the inlet, where

$$u = \frac{4}{H^2}(H - y)y \tag{9}$$

$$v = 0 \tag{10}$$

The no-slip wall boundary condition is enforced at the particle's edge as well as at the top and bottom boundaries of the domain. At the right end of the domain, a zero-pressure outlet condition is applied. This framework is tested at particle *Re* of 5, 20, and 50. To create variations in the Reynolds number, an imaginary fluid is created, maintaining a constant density of 1 kg/m$^3$ across all cases, while the dynamic viscosity is set at 0.2, 0.05, and 0.02 Pa·s, respectively. The viscous heat is neglected in this study.

There are 400 by 200 sample points evenly distributed in the entire domain, with 400 in the x direction and 200 in the y direction. The region near the particle is refined with more sample points. In total, there are approximately 100 thousand sample points in the domain. Figure 3 shows the sample points of the domain.

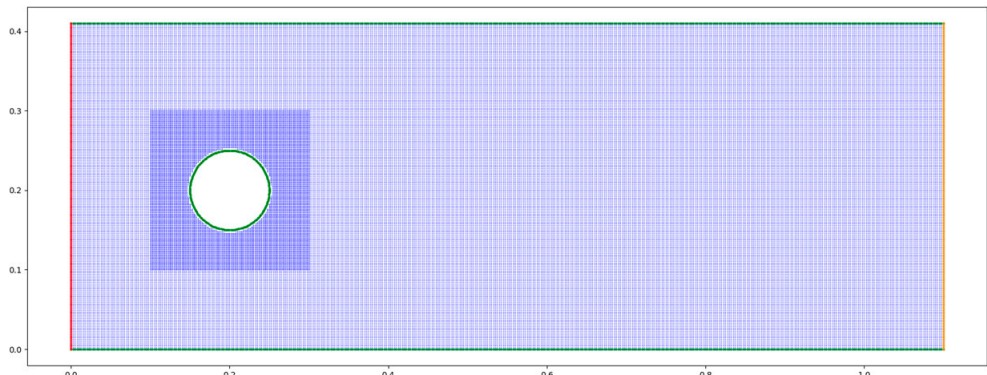

**Figure 3.** Sample points in the domain. Red points are the inlet, green points are the wall, blue points are the interior points, and yellow points are the outlet.

*2.5. Drag Force Coefficient*

The particle drag force coefficient is calculated as a numerical metric to quantify the error between the PINNs and CFD results. The particle drag coefficient $C_d$ is defined as Equation (11):

$$F_d = \frac{1}{2} C_d \rho v^2 A_{\text{frontal}} \tag{11}$$

where $F_d$ represents the drag force applied to the particle in the streamwise direction, which, in this instance, corresponds to the x-direction. The drag force consists of two components in the low-Re flow regime, namely pressure-driven drag force and viscous drag force, as graphically depicted in Figure 4, where $r$ is the radius of the circular particle and $\vec{n}$ is the unit normal vector of the particle. In conventional CFD using Finite Volume Method, the fluid-solid interaction force can be easily obtained by calculating the total force exerted on all surrounding fluid grids adjacent to the particle wall, as shown in Equation (12):

$$F_d = \sum \left[ -\frac{\partial p}{\partial x} + \mu \left( \frac{\partial^2 u}{\partial x^2} + \frac{\partial^2 u}{\partial y^2} \right) \right] V_{\text{cell}} \tag{12}$$

where $V_{\text{cell}}$ is the volume of each grid adjacent to the particle wall. However, due to the nature of the PINNs, which utilize a point-based system, this methodology cannot be applied as there is no fluid grid volume. As a result, the drag force is determined by numerically integrating the pressure $p$ and the stress tensor $\tau$ on the surface of the particle.

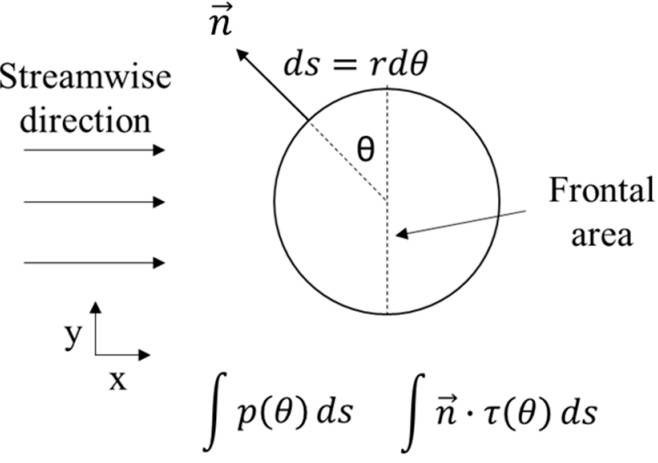

**Figure 4.** Illustration of the drag force calculation.

- **Pressure drag force**

    Assuming the pressure and stress tensor on the wall are a function of the angle on the circular particle, the pressure force is equal to the integration of the pressure over the surface of the particle on the upstream side (left to the frontal area) minus the downstream side (right to the frontal area), as shown in Equation (13).

$$F_{dp} = \int p(\theta) r d\theta_{\text{up}} - \int p(\theta) r d\theta_{\text{down}} \tag{13}$$

Based on the trapezoidal rule for integration, the above integration can be discretized as in Equation (14).

$$\int p(\theta) r d\theta = \sum_{i=0}^{n} \frac{1}{2} (p(\theta_i) + p(\theta_{i+1})) r \Delta\theta \tag{14}$$

where $n$ is the number of sample points sampled on the particle wall. If the sample points are evenly distributed on the entire surface of the particle, then $\Delta\theta$ is a constant and is equal to $\pi$ divided by the number of points along the circumference of the particle. Then Equation (14) can be simplified, and the pressure drag force is equal to $\pi r$ times the arithmetic mean of the pressure at all points, as shown in Equation (15). The pressure force can be further written in discrete form, as shown in Equation (16):

$$\sum_{i=0}^{n} \frac{1}{2} (p(\theta_i) + p(\theta_{i+1})) r \frac{\pi}{n} = \pi r \frac{\sum_{i=0}^{n} p(\theta_i)}{n} = \pi r \overline{p} \tag{15}$$

$$F_{dp} = \pi r (\overline{p_{\text{up}}} - \overline{p_{\text{down}}}) \tag{16}$$

where $\overline{p_{\text{up}}}$ is the arithmetic mean of the pressure at all sample points in the upstream side of the particle, and $\overline{p_{\text{down}}}$ is the arithmetic mean of the pressure at all sample points in the downstream side of the particle.

- **Viscous drag force**

    The viscous force is calculated by integrating the unit normal vector of the given surface over the entire surface of the particle times the stress tensor $\tau = \begin{bmatrix} \tau_{xx} & \tau_{xy} \\ \tau_{yx} & \tau_{yy} \end{bmatrix}$, as written in Equation (17):

$$F_s = \int \vec{n} \cdot \tau(\theta) r d\theta \tag{17}$$

The unit normal vector at any point of the circular particle can be written as $[-\sin\theta, \cos\theta]$. Therefore, the viscous force can be written as shown in Equation (18).

$$F_s = \int \left[ -\tau_{xx}\sin\theta + \tau_{yx}\cos\theta \quad -\tau_{xy}\sin\theta + \tau_{yy}\cos\theta \right] r d\theta \tag{18}$$

As the drag force is only in the streamwise direction ($x$ direction), the drag force due to the viscous stress is therefore written as

$$F_{ds} = \int \left( -\tau_{xx}\sin\theta + \tau_{yx}\cos\theta \right) r d\theta. \tag{19}$$

For a Newtonian fluid, the viscous stresses are proportional to the element strain rates and the viscosity. $\tau_{xx} = 2\mu \frac{\partial u}{\partial x}$, $\tau_{yx} = \mu \left( \frac{\partial u}{\partial y} + \frac{\partial v}{\partial x} \right)$. Therefore, the drag force can be written as

$$F_{ds} = \int \left[ -2\mu \frac{\partial u}{\partial x} \sin\theta + \mu \left( \frac{\partial u}{\partial y} + \frac{\partial v}{\partial x} \right) \cos\theta \right] r d\theta. \tag{20}$$

Similar to the pressure force, after applying the trapezoidal rule for integration, the above integration can be discretized as in Equation (21) below.

$$F_{ds} = \sum_{i=1}^{n} \left[ -2\mu \frac{\partial u}{\partial x} \sin\theta_i + \mu \left( \frac{\partial u}{\partial y} + \frac{\partial v}{\partial x} \right) \cos\theta_i \right] \frac{2\pi r}{n} \tag{21}$$

### 2.6. CFD Validation

CFD simulations were conducted to validate the results of the proposed PINNs framework using ANSYS Fluent 2020R2. A grid with a comparable resolution of hexagonal cells was configured for the CFD simulation, as illustrated in Figure 5. The grid consists of 400 cells spanning the horizontal section of the domain and 200 cells covering the vertical segment. An additional five layers of inflation cells were integrated adjacent to the wall, resulting in 77,414 cells in total. The same parabolic velocity profile in Equations (10) and (11) is applied as an inlet boundary condition. A zero-pressure boundary condition is applied to the outlet. No-slip wall conditions are applied to the surface of the particle and the top and bottom of the domain. The SIMPLE scheme is applied for pressure-velocity coupling, and second-order upwind discretization is used for pressure and momentum terms. Convergence is reached when the relative residual of all terms is below $10^{-5}$. The CFD simulation is performed on an Intel i9-10940X at 3.3 Ghz.

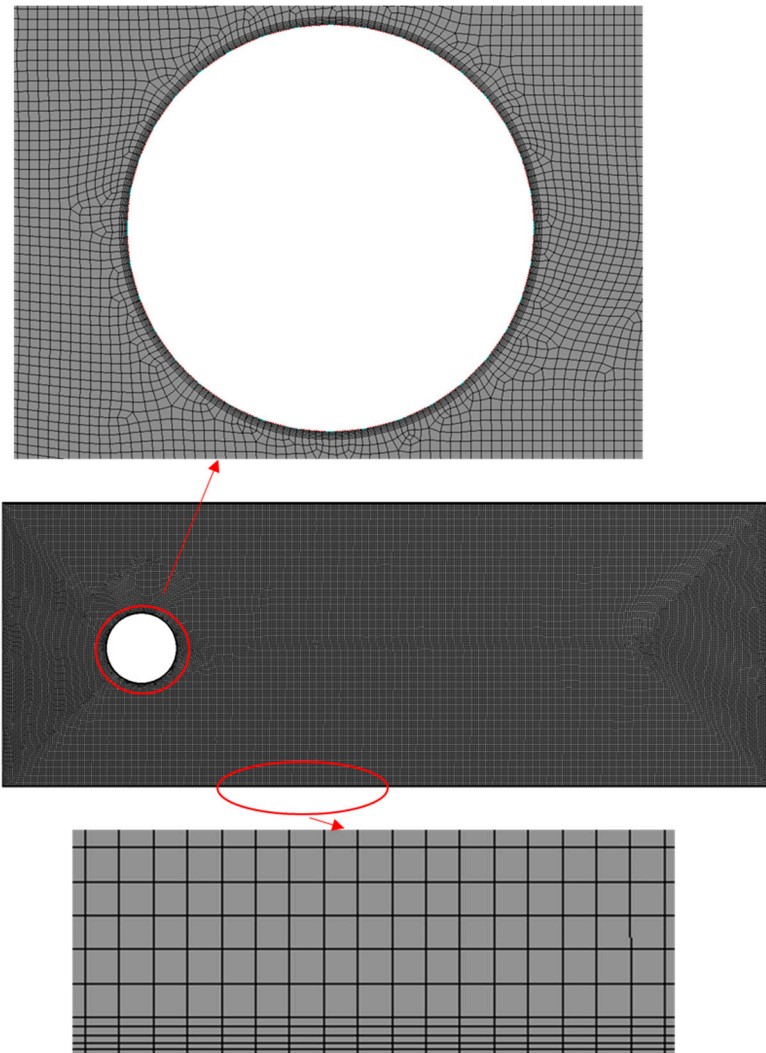

**Figure 5.** Computational grid of the benchmark case in CFD.

## 3. Results

Figure 6 shows the comparison between the proposed framework and its CFD validations. Velocity contours and vectors, along with pressure contours, are displayed at $Re$ = 5, 20, 50, as shown in Figures 6–8. Overall, the PINNs framework effectively solves the Navier–Stokes equations and identifies the unique solution considering all the boundary conditions within the specified domain.

In all three cases, the PINNs framework manages to capture the flow characteristics found in the stagnation region, flow separation region, venturi region, and wake region. In the stagnation zone located at the forefront of the particle, the fluid velocity reduces to zero. In the pressure contour, the stagnation zone is marked by a small high-pressure zone upstream of the particle due to the conservation of energy. The fluid then divides into two streams that pass around the particle. Due to a reduction in flow area, the fluid accelerates both above and below the particle, a phenomenon also referred to as the "venturi effect". Concurrently, the pressure rapidly declines within this region. Subsequently, the two streams of fluid reunite, forming a wake downstream of the particle. Given that the Reynolds number is low in all three instances, the wake presents less chaotic behavior. Nonetheless, a flow separation followed by a region of recirculating fluid behind the particle can still be observed in the vectors. In the pressure contour, the flow separation is indicated by a near-zero pressure zone situated behind the particle. The adverse pressure gradient, where the pressure gradually increases in the streamwise direction, can also be observed in the flow separation zone. In the velocity contour of all three cases, clear stratification can be observed near the wall region, representing the boundary layer of the wall.

At $Re$ = 5, the PINNs framework accurately captures the maximum velocity both above and below the particle due to the "venturi effect". A slightly larger discrepancy can be observed in the downstream wake region. The wake in the PINNs is slightly wider than in the CFD. In the pressure contour, despite some discrepancies near the bottom left corner, the overall pressure pattern is well captured. At $Re$ = 20 and 50, the wake is broadened due to the increase in $Re$. While the PINNs faithfully capture the elongated wake region and the flow separation, it shows a slightly lower maximum velocity compared to the CFD results, suggesting a slight mass imbalance in the PINNs result. Furthermore, the discrepancy in pressure also increases as the $Re$ increases.

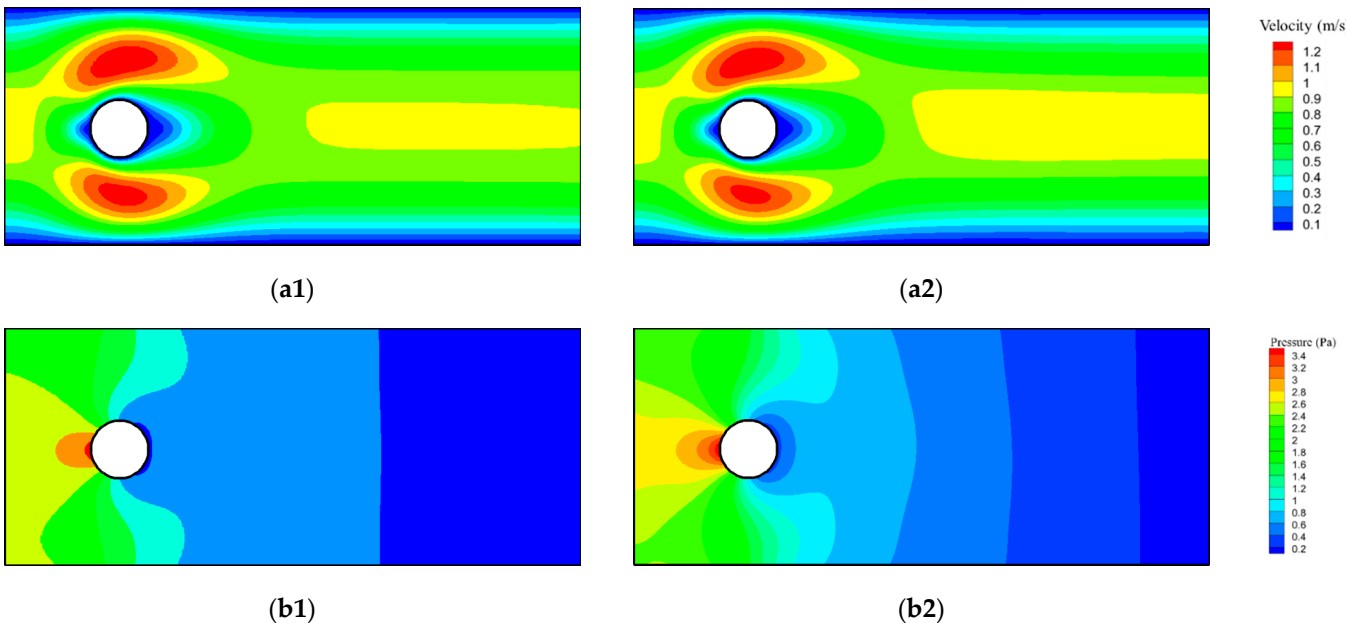

(a1)                    (a2)

(b1)                    (b2)

**Figure 6.** *Cont.*

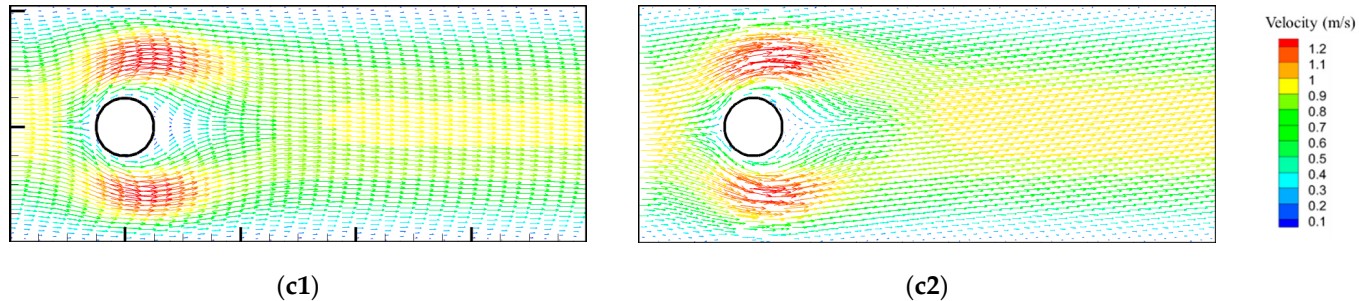

(**c1**)                                    (**c2**)

**Figure 6.** Result comparisons of CFD and PINNs of flow passing around a 2D circular particle at $Re = 5$. (**a1**) Velocity contour of CFD at $Re = 5$; (**a2**) velocity contour of PINNs at $Re = 5$; (**b1**) pressure contour of CFD at $Re = 5$; (**b2**) pressure contour of PINNs at $Re = 5$; (**c1**) velocity vectors of CFD at $Re = 5$; (**c2**) velocity vectors of PINNs at $Re = 5$.

(**a1**)                                    (**a2**)

(**b1**)                                    (**b2**)

(**c1**)                                    (**c2**)

**Figure 7.** Result comparisons of CFD and PINNs of flow passing around a 2D circular particle at $Re = 20$. (**a1**) Velocity contour of CFD at $Re = 20$; (**a2**) velocity contour of PINNs at $Re = 20$; (**b1**) pressure contour of CFD at $Re = 20$; (**b2**) pressure contour of PINNs at $Re = 20$; (**c1**) velocity vectors of CFD at $Re = 20$; (**c2**) velocity vectors of PINNs at $Re = 20$.

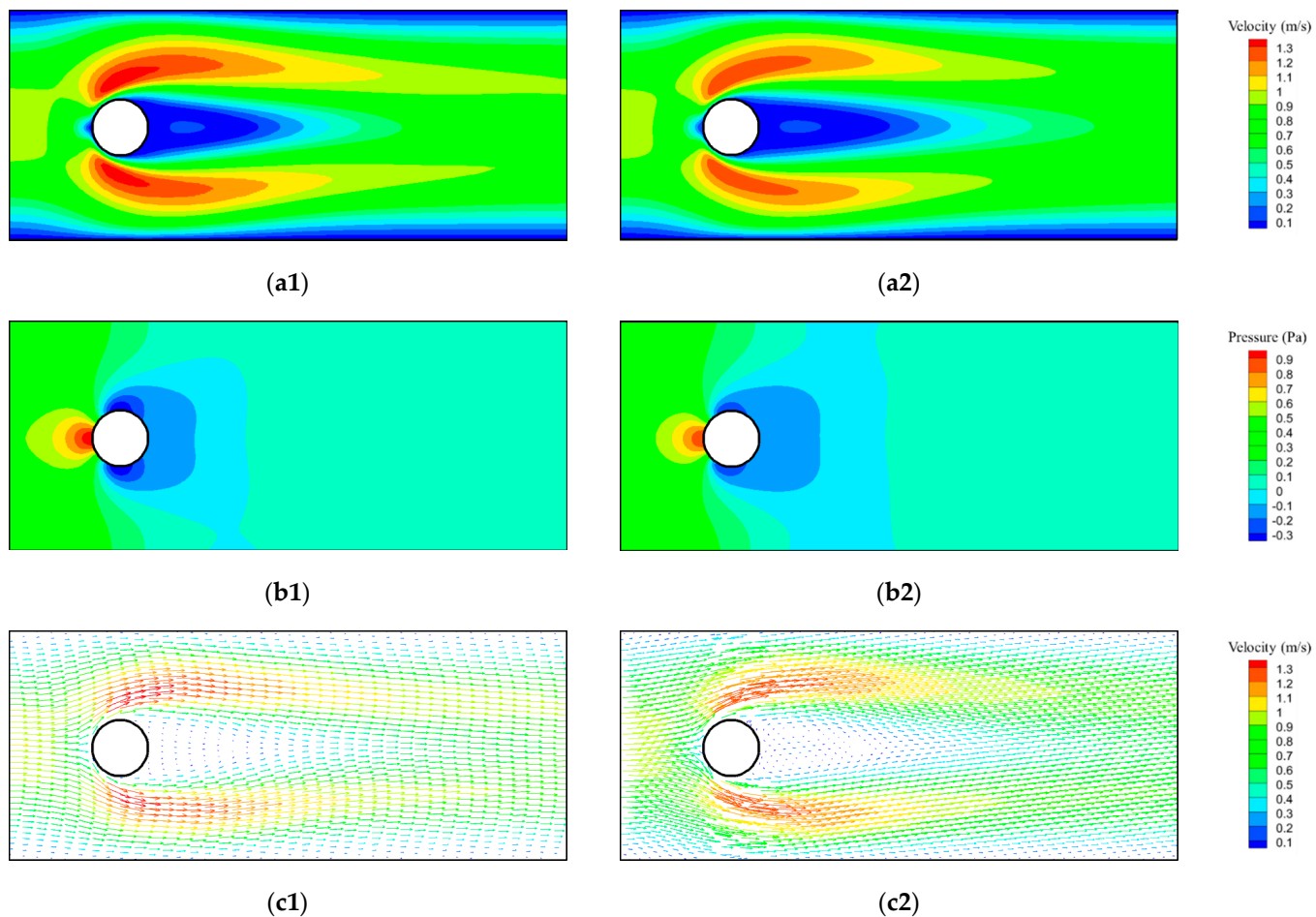

**Figure 8.** Result comparisons of CFD and PINNs of flow passing around a 2D circular particle at
*Re* = 50. (**a1**) Velocity contour of CFD at *Re* = 50; (**a2**) velocity contour of PINNs at *Re* = 50;
(**b1**) pressure contour of CFD at *Re* = 50; (**b2**) pressure contour of PINNs at *Re* = 50; (**c1**) velocity vectors of CFD at *Re* = 50; (**c2**) velocity vectors of PINNs at *Re* = 50.

This discrepancy can be attributed to insufficient convergence, which is thought to
be related to the way boundary conditions are enforced in PINNs. In CFD, boundary
conditions are strictly specified; scalar variables such as velocity and pressure in boundary
cells are used directly to calculate the interior cells adjacent to the boundary. Consequently,
there are no residuals for boundary conditions in CFD. However, in the proposed PINN
framework, boundary conditions are softly enforced, being incorporated as part of the
loss function. The residual on the boundary points is aimed at being zero, but achieving
this goal depends heavily on the training process (optimization of the loss function) of
the PINNs. To illustrate, consider the non-slip wall boundary condition. Even though
the velocity on the wall is specified as zero, the residual at the wall boundary points after
training might be non-zero, even though it might be minimal. When a sufficient number
of wall boundary points exhibit a small residual, the overall mass of the domain may not
be conserved. Consequently, at certain locations on the wall, the fluid might demonstrate
a small velocity that points outward from the wall. Furthermore, the neural network is
designed to generalize predictions and avoid overfitting, meaning it does not strive to
perfectly match the target residual at every single sample point within the domain. Given
that boundary points constitute approximately 5% of the total sample points, the training
is unlikely to progress further, even if a minimal residual persists at the boundary points.

The drag force coefficients of the particle are calculated through the finite differencing
method described in Section 2.5 to quantify the error of the PINNs compared to the CFD,
as shown in Table 1. Given that the flow is in the laminar regime, viscous forces play a

significant role and are comparable to inertial forces. Therefore, both the viscous drag coefficient and the pressure drag coefficient are calculated, and the total drag coefficient is the sum of these two terms. When compared to the CFD results, both the pressure and viscous drag coefficients computed from the PINNs across all cases are found to be less than 10%, with the error in the total drag coefficient less than 6%. Overall, the drag coefficient predictions made by PINNs show reasonable agreement with the CFD results.

**Table 1.** Particle drag coefficient in PINNs and CFD.

|  | Viscous Drag Coeff. | | Pressure Drag Coeff. | | Total Drag Coeff. | |
|---|---|---|---|---|---|---|
|  | CFD | PINNs | CFD | PINNs | CFD | PINNs |
| *Re* = 5 | 3.78 | 3.34 | 5.09 | 5.53 | 8.87 | 8.87 |
| *Re* = 20 | 1.19 | 1.06 | 1.92 | 1.86 | 3.12 | 2.92 |
| *Re* = 50 | 0.61 | 0.55 | 1.32 | 1.28 | 1.94 | 1.83 |

The flow passing around an elliptical particle is also tested with the same PINN framework. Figure 9 shows the result of flow passing around a 2D elliptical particle at *Re* = 20. Similar to the circular cylinder case, the PINNs framework captures most of the flow features observed in the stagnation region, flow separation region, venturi region, and wake region. Given the elongated and narrower shape of the particle, the flow separation region is extended as well. However, the maximum velocity in both the venturi region and the wake region is slightly underpredicted by the PINN framework. Aside from this, the PINN framework accurately captures the flow field dynamics.

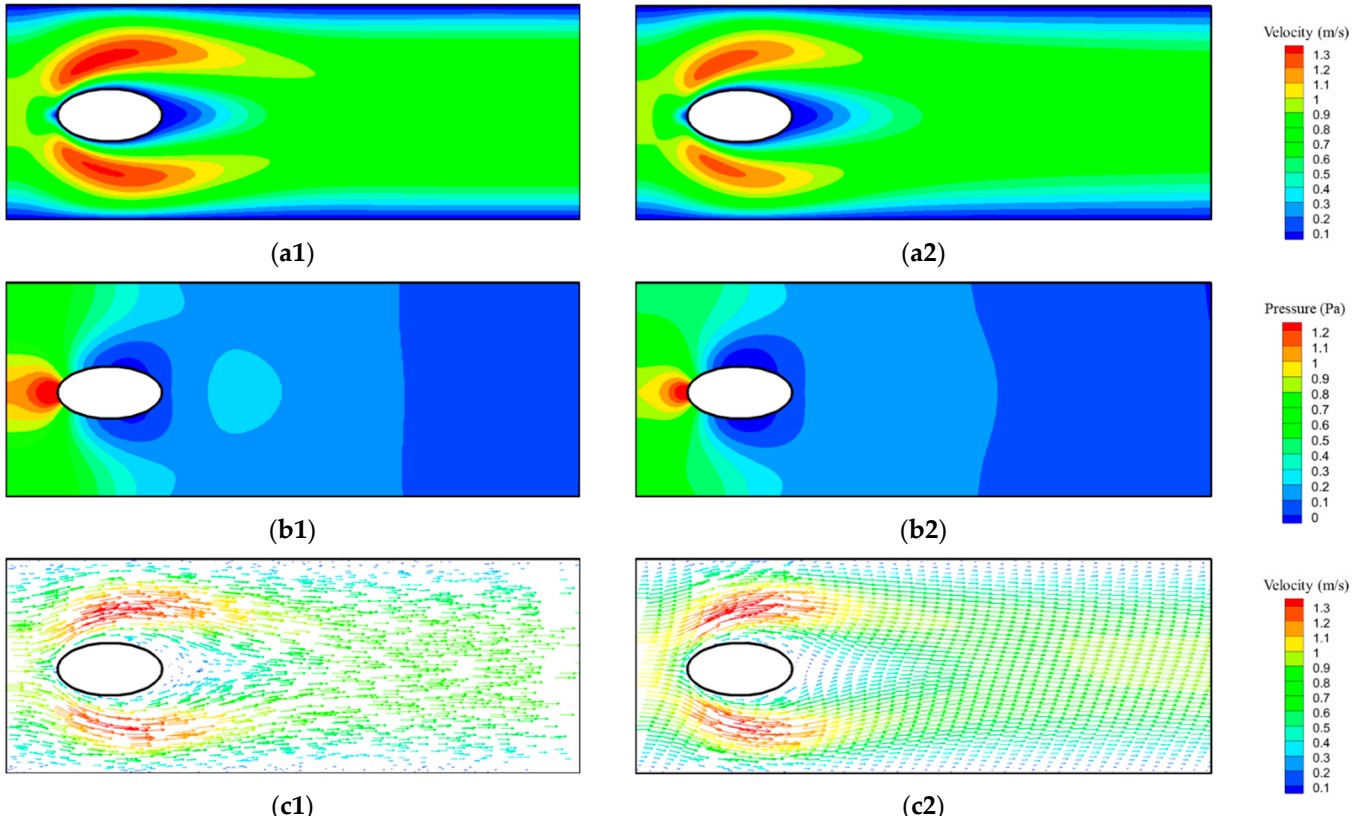

**Figure 9.** Result comparisons of CFD and PINNs of flow passing around a 2D elliptical particle at *Re* = 20. (**a1**) Velocity contour of CFD; (**a2**) velocity contour of PINNs; (**b1**) pressure contour of CFD; (**b2**) pressure contour of PINNs; (**c1**) velocity vectors of CFD; (**c2**) velocity vectors of PINNs.

Figure 10 shows the training history of the proposed PINN framework for the case of circular particles at *Re* = 5. The training history outlines the loss function's residual over the number of epochs. The depicted residual behaves similarly to those observed in common CFD simulations. The total residual is the sum of the residuals of the continuity equation, x-momentum, and y-momentum. It can be observed that the initial convergence is slow, with the residual beginning to decrease significantly after 6000 iterations. The final convergence takes place over 10,000 epochs. A spike observed in the training might be attributed to the adaptive learning rate of the Adam optimizer, which can occasionally cause sudden jumps in the parameter space, resulting in spikes in the loss function. Subsequently, as the training algorithm transitions to L-BFGS following 10,000 iterations, the training demonstrates increased stability. The training stops when the gradient of each term is less than $1 \times 10^{-7}$, around 140,000 iterations.

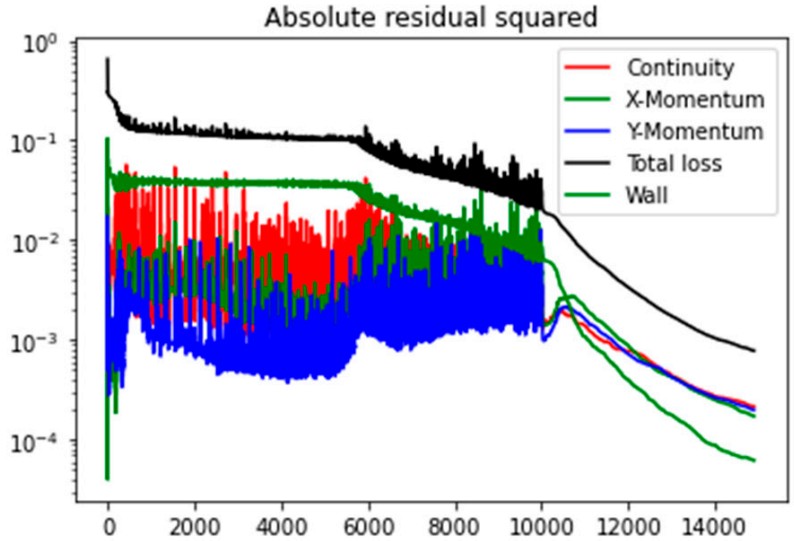

**Figure 10.** Training history of the case *Re* = 5. Residual vs. iterations.

The training time and hardware are listed in Table 2. The inference time for the PINN to predict the velocity and pressure across all sample points in the domain is 0.07 s. It should be noted that deep learning processes are typically performed on a GPU, while CFD simulations are usually conducted on a CPU. Although this does not constitute a direct comparison, the time efficiency of PINNs remains unrivaled by CFD, even when trained on a GPU. However, one positive aspect to consider is that PINNs do not require a computation grid, eliminating the often time-consuming task of generating a fine mesh that is required in many real-world applications.

**Table 2.** Training time in PINNs and CFD.

| Hardware | CFD | PINNs |
|----------|-----|-------|
| Intel i9-10940X 3.3 Ghz (CPU) | 2 min | 16 h |
| Nvidia Tesla P100-16 GB (GPU) | / | 1.5 h |

## 4. Conclusions

A physics-informed neural network to solve steady-state, 2D incompressible Navier–Stokes is proposed. The framework has been tested on flows passing around a circular and elliptical particle at *Re* numbers. The PINNs effectively capture most flow features, with the error in the predicted drag coefficient being less than 10% in all cases compared to CFD results. It should be noted that the current PINN framework is not intended to replace traditional CFD methods, as it exhibits a longer training time and lower accuracy. However, it serves as a proof of concept, presenting a novel approach to

solving Navier–Stokes equations. Future work will focus on enhancing the enforcement of boundary conditions and reducing training time. In addition, future work will utilize the generalization capabilities of neural networks by incorporating different boundary conditions into training as a parameter so that the PINN framework can predict the flow field with different boundary conditions.

**Author Contributions:** Conceptualization, B.H.; methodology, B.H.; software, B.H.; validation, B.H.; formal analysis, B.H.; investigation, B.H.; resources, D.M.; data curation, B.H.; writing—original draft preparation, B.H.; writing—review and editing, B.H. and D.M.; visualization, B.H.; supervision, D.M.; project administration, D.M.; funding acquisition, D.M. All authors have read and agreed to the published version of the manuscript.

**Funding:** This research was funded by the Department of Energy, grant number DE-FE0031904.

**Data Availability Statement:** No new data were created or analyzed in this study. Data sharing is not applicable to this article.

**Acknowledgments:** We would like to express our deepest gratitude to Cheng-Xian Lin, who supervised this research but, regrettably, passed away before its completion. His guidance and expertise were invaluable in shaping this study.

**Conflicts of Interest:** The authors declare no conflict of interest.

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
