# Peer review of "Applying Physics-Informed Neural Networks to Solve Navier–Stokes Equations for Laminar Flow around a Particle"

_mca, doi:10.3390/mca28050102_

Round 1

Reviewer 1 Report

This manuscript evaluates a physics-informed neural network (PINN) for the Navier-Stokes equations (NSE), for the case of 2D flow around a particle.  PINNs have been suggested for the NSE before, and although the authors list related research, they do not put it in perspective with the current work. The NSFnet [12] does for example seem similar to the net in this paper, and that paper also studies 2D flow around a cylinder. The authors need to make it clearer what the actual contributions of this manuscript are.

It should also be discussed whether this is a reasonable way of solving the NSE. The neural network here is physics-informed, but otherwise solving the problem in a very unstructured way. The method is essentially fitting a global, highly nonlinear, approximant to the PDE, by enforcing the equations at a number of collocation points. How does this compare to e.g. a spectral method with global basis functions? This especially important since the proposed methods takes 16 hours to run on a CPU, while the reference algorithm runs in only 2 minutes.

Another related question is how does this make use of the generalization capabilities of neural networks? As far as I understand it, the method is "trained" to solve a specific problem on a specific domain, and cannot generalize in any way. 

More comments, going from top to bottom in manuscript:

Line 15-16: The problem is in 2D, so the particles is circular and elliptical, not spherical and ellipsoidal.

Line 33: Is it only due to new training algorithms?

Section 2.3: The neural network architecture is very simple, consisting of 8 fully connected layers, each 60 wide. Is it possible to use more advanced architectures, that somehow encode knowledge about the problem at hand?

Related, how many trainable degrees of freedom does the network have?

Equation 6: Should there be a square root on the square of the norm?

Line 144: The loss function will not be zero, so the predicted solution approximates the equations (doesnt adhere to them).

Line 158: Why the specific value alpha=2?

Line 162: Will you publish your open source?

Line 175: You claim that a benefit of your method is that you do not need to mesh, yet you manually add refinement points near the particle. Is this not a problem?

It would also be nice to have a plot of the sample points.

Figure 4: Very low resolution of the image, so it is hard to see the grid.

Discussion in section 3 talks very much about what the solution looks like. it would be more informative to actually plots the errors, eg |u_{ref}-u_{PINN}|, for the various quantities and flow cases.

Figure 9: It looks like you stopped training while it was still converging. Why is that?

Table 2: What is the inference time of the trained network, when evaluating it at all your points?

How does the number of degrees of freedom compare between the two methods?

Some proof-reading required. PINNs is plural, but the singular form should probably be used when discussing the particular network of this work.

Reviewer 2 Report

See attached.

refer to attached
